# Conditional Protein Sequence Generation Using Protein Language Models with Adapters

## Abstract

The conditional generation of proteins with desired functions and/or properties is a key goal for generative models. Existing methods based on prompting of language models can generate proteins conditioned on a target functionality, such as a desired enzyme family. However, these methods are limited to simple, tokenized conditioning and have not been shown to generalize to unseen functions. In this study, we propose ProCALM (**Pro**tein **C**onditionally **A**dapted **L**anguage **M**odel), an approach for the conditional generation of proteins using adapters to protein language models (PLMs). While previous methods have used adapters for structure-conditioned generation from PLMs, our implementation of ProCALM involves finetuning ProGen2 to condition generation on protein function, and it is flexible to function representations of enzyme family, taxonomy, or natural language descriptions. ProCALM matches or exceeds the performance of existing methods at conditional sequence generation from target functions. Impressively, it can also generalize to rare and unseen functions. Overall, ProCALM is a flexible and computationally efficient approach, and we expect that it can be extended to a wide range of generative language models.

## 1 Introduction

Proteins, sequences of amino acids, are important molecules in all living organisms and have many industrial applications. Protein sequences can be modified or designed to have desired function(s) or optimized properties so that they are more useful for applications ranging from greener chemical synthesis to gene-editing for disease treatment (Buller et al., 2023). Enzymes are a particularly useful subclass of proteins: these are ubiquitous proteins that catalyze chemical reactions and are particularly difficult to engineer, as enzymatic activities are complex and often difficult to predict from sequence (Arnold, 2018).

While directed evolution and other methods have been used to engineer proteins (Wang et al., 2021), in recent years, generative models based on machine learning (ML) have emerged to tackle protein design (Ruffolo & Madani, 2024; Wu et al., 2021; Xie & Warshel, 2023; Barghout et al., 2023; Ferruz et al., 2023). The use of generative models is motivated by learning the distribution of known proteins, thus allowing one to sample functional proteins during inference. Diffusion models can generate protein structures that have certain symmetries, binding properties, or active site conformations (Watson et al., 2023; Ingraham et al., 2023; Huguet et al., 2024; Lauko et al., 2024). Discrete diffusion (Alamdari et al., 2023; Wang et al., 2024a), Bayesian flow models (Atkinson et al., 2024), and other approaches (Yang et al., 2024a) are effective generative models for sequences, but to date, protein language models (PLMs) based on autoregressive transformers have demonstrated the most success for generating functional enzymes (Madani et al., 2023; Nijkamp et al., 2023; Ferruz et al., 2022; Zvyagin et al., 2022; Hesslow et al., 2022).

An important downstream application for generative models is the conditional generation of proteins, where a "condition" is typically defined by desired function and/or properties (Yang et al., 2024a; Ferruz & Höcker, 2022; Dai et al., 2024; Notin et al., 2024; Hayes et al., 2024). For example, it is desirable to design Cas9 nucleases that express well and have high activity and specificity for gene-editing (Chen & Liu, 2023). Alternatively, enzyme engineers might be interested in thermostable enzymes that can catalyze unannotated, non-native reactions for chemical synthesis (Chen & Arnold, 2020), among numerous other applications (Buller et al., 2023). Conditional generation with ML

models to maximize a desired property such as stability has been explored using direct preference optimization and guided diffusion (Widatalla et al., 2024; Nisonoff et al., 2024; Gruver et al., 2023; Lisanza et al., 2024). Here, we focus on methods for generating sequences with a target function, which is more qualitative rather than quantitative.

A typical approach for generating sequences with a target function, such as specific enzymatic activity, involves finetuning a PLM on a specific class or family of enzymes and then generating novel enzymes with that function (Madani et al., 2023; Ruffolo et al., 2024b) – an approach we will refer to as *Target Finetuning*. This approach has been successfully applied to ProGen and other similar models with wet-lab validation (Nicolini et al., 2024; Zvyagin et al., 2022; Munsamy et al., 2024). Other successful recent approaches for conditional protein sequence generation (such as ZymCTRL) are related to *Prompting* (Munsamy et al., 2024; Luo et al., 2023; Nathansen et al., 2023; Liu et al., 2024). Despite these advances, a major limitation of existing models is that they can only be finetuned for known function. This requires extensive prior knowledge of a protein family with that function (Table 1), but protein engineers are often interested in new functions, such as enzymes with new-to-nature activities (Arnold, 2018). Thus, current methods are not flexible to complex conditioning (such as jointly across enzymatic function and organism) and may not generalize well to functions that lie out-of-distribution (OOD).

To address this challenge, our **key contribution** in this study is introducing a broadly applicable strategy (*Conditional Adapters*) for parameter efficient finetuning of language models. While previous approaches using adapters have been largely limited to protein structure conditioning, we show that *Conditional Adapters* are flexible and useful for conditioning generation from PLMs based on different types of desired functions. Specifically, we apply this approach to finetune ProGen2 to conditionally generate protein sequences based on enzyme family, taxonomy, and natural language descriptions – called ProCALM (**Pro**tein **C**onditionally **A**dapted **L**anguage **M**odel). Our approach demonstrates several advantages compared to existing methods:

1. ProCALM is parameter efficient and computationally inexpensive to train.
2. ProCALM is a general approach that is flexible to various types of conditioning (beyond just structure) – including enzyme class, taxonomy, and textual descriptions.
3. ProCALM matches or surpasses existing methods at generating protein sequences with target functions.
4. ProCALM can generalize toward conditions that lie OOD, namely by generating sequences for combinations of taxonomy and enzyme class that are not seen in the training set.

## 2 RELATED WORK

**Prompting and Related Approaches.** While many approaches have been explored for steerable generation from language models, the most common approaches involve using prompts, or tokens at the beginning of a sequence, to guide the generation of the remaining sequence (*Prompting*). For example, in the CTRL transformer architecture, control tags are used to guide natural language generation towards specific styles – such as science vs politics (Keskar et al., 2019). Munsamy et al. (2024) trained a PLM based on the CTRL transformer architecture, where the control tags were enzyme commission (EC) numbers describing enzyme families (ZymCTRL). Many of the protein sequences generated from ZymCTRL were successfully validated in the wet-lab as functional enzymes.

While CTRL-type models are trained from scratch, prompt tuning and related parameter-efficient approaches involve finetuning pretrained language models using prompts (Lester et al., 2021; Li & Liang, 2021; Zeldes et al., 2020). The weights associated with the pretrained model are fixed, and only the weights associated with the new tokens are trained. There are a few examples of prompt tuning being applied to protein generation. In PrefixProt, ProtGPT2 was finetuned to generate antimicrobial peptides enriched in alpha helices (Luo et al., 2023). In Finenzyme, ProGen was finetuned using prefix tuning on seven broad EC classes, and a few selected specific ECs (Nicolini et al., 2024). Nathansen et al. (2023) explored this strategy for finetuning RITA to generate a specific protein family.

Another interesting way to condition for function is through natural language. PROPEND uses prompt tuning to finetune PLMs for conditional generation based on backbone, secondary structure,

Table 1: **Using conditional adapters offers several advantages for the conditional generation of proteins using language models.** It is inherently more flexible for different types of conditioning (including multiple functions); it is more generalizable to out-of-distribution (OOD) conditions; and it is computationally efficient to train. A visualization of various approaches for conditional generation is provided in Fig. 1C.

| Approach | Flexibility | OOD | Cost | PLM Example(s) |
|---|---|---|---|---|
| *Target Fine-tuning* | Restricted to a target condition | Cannot generalize OOD | Expensive if scaled | applied post-training |
| *Prompting* | Largely restricted to a tokenized condition | Generalizes weakly OOD | Expensive if trained from scratch | ProGen (Madani et al., 2023), ZymCTRL (Munsamy et al., 2024), Finenzyme (Nicolini et al., 2024), PrefixProt (Luo et al., 2023), PROPEND (Wang et al., 2024b), ProteinDT (Liu et al., 2024) |
| *Conditional Adapters* | Flexible to multiple conditions | Could generalize OOD | Parameter efficient | LM-Design (Zheng et al., 2023), ShapeProt (Lee & Kim, 2024), InstructPLM (Qiu et al., 2024), proseLM (Ruffolo et al., 2024a), **ProCALM** (our study) |

and natural language (Wang et al., 2024b). Many other text-guided conditional generative models of protein sequences take inspiration from classifier-free guidance in image generation (Ramesh et al., 2022). For example, a CLIP-like representation is learned between text and protein sequences, which is then decoded downstream using a diffusion model (Liu et al., 2024; Praljak et al., 2024) or an autoregressive transformer (Liu et al., 2024; Yin et al., 2024). In this work, we compare against ProteinDT, which uses an indirect form of prompt tuning to finetune PLMs for conditional generation based on "CLIP" representations of protein and natural language textual descriptions (Liu et al., 2024).

Overall, *Prompting* and related approaches can sometimes work well for conditional generation when the prompt is within distribution. Still, ZymCTRL was trained from scratch. Other studies from natural language processing suggest that *Prompting* is not always the best approach (Chen et al., 2022). In the past protein studies listed above, a major limitation of such approaches is that they do not enable (or haven't been tested for) generation over a broad range of specific functions. Furthermore, these approaches are restricted to using simple conditions that can be tokenized and have not been studied for the generation of sequences for conditions that lie OOD (Table 1).

**Adapters in Language Models.** Adapter-based tuning of language models has emerged as a parameter-efficient strategy (Pfeiffer et al., 2021; Hu et al., 2021) to finetune language models for specific tasks (Ribeiro et al., 2021; Houlsby et al., 2019), such as describing protein sequences with natural language (Carrami & Sharifzadeh, 2024; Huo et al., 2024). Recently, using *Conditional Adapters* has shown utility in PLMs for the conditional generation of proteins. Several studies have used adapters to condition generation from PLMs based on desired structures (Zheng et al., 2023; Lee & Kim, 2024; Qiu et al., 2024). In proseLM, generation was conditioned on protein structure (Ruffolo et al., 2024a), and generated gene editing enzymes and antibodies that were successfully validated in the wet lab. LM-Design is particularly interesting because the authors also evaluated generalization toward unseen protein folds (Zheng et al., 2023). Conditioning on structure is useful, as structure often determines function, but a goal of protein engineering is often to find proteins with novel functions (such as for new-to-nature enzyme activity (Arnold, 2018)), without any known structure or sequence performing this function. Thus, there is a need to explore models that can condition directly on function and generalize to new functions.

Overall, using *Conditional Adapters* is promising and offers several advantages (Table 1, Fig. 1). Most importantly, the approach has greater potential to generate sequences beyond the training distribution because conditioning is performed in continuous space, rather than through initial tokenization like *Prompting*. Still, a few key questions remain, which we aim to explore in this study. (1) Are *Conditional Adapters* amenable to different types of conditioning information for protein generation (i.e. beyond structure)? (2) How does the quality of generated sequences from such a model compare

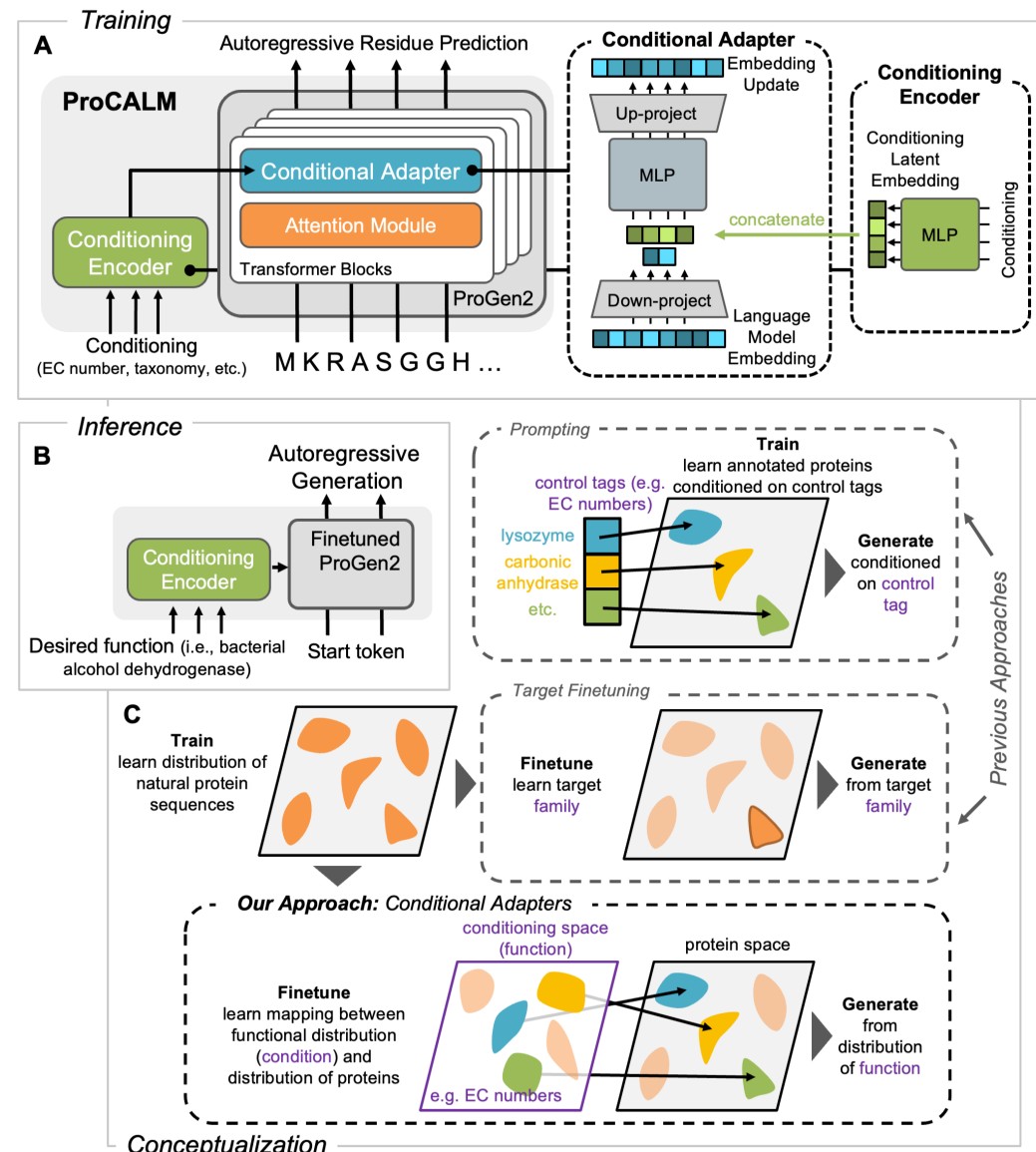

Figure 1: **Finetuning PLMs with conditional adapters is a flexible approach for the conditional generation of proteins with desired function.** **(A)** Training of ProCALM involves finetuning ProGen2 in a parameter efficient manner. **(B)** During inference, sequences are generated autoregressively, conditioned on a representation encoding desired function. **(C)** Conceptualization of our approach in the context of previous approaches. Conditioning is flexible, and the model captures a notion of similarity between the desired conditioning and conditions from the training set. MLP stands for multi-layer perceptron.

to existing approaches? (3) To what extent does the approach enable generation of sequences from conditions that lie OOD?

## 3   OVERVIEW OF PROCALM

ProCALM is trained via parameter-efficient finetuning of ProGen2 (Nijkamp et al., 2023) with conditional adapters; the original ProGen2 weights are fixed while the conditional adapter layers and conditioning encoder are trained (Fig. 1A). The conditioning encoder learns a latent representation of conditioning information, which is concatenated with a low-rank projection of the language model hidden embedding inside each conditional adapter layer (corresponding to each transformer layer).

Table 2: **Summary of datasets and splits used for training and evaluation of ProCALM on EC-related tasks.** Unless otherwise noted, models were trained on the *Swissprot Train* set using the ProGen2-Base model (764 million parameters). For the text-guided generation task, the same training dataset as ProteinDT was used, and no dataset resampling or splitting was performed.

| Name | Purpose | Description | Sequences | ECs |
|------|---------|-------------|-----------|-----|
| *Uniref* | Training | Sequences in Uniref associated with EC numbers | $29.4 \times 10^6$ | 5222 |
| *Swissprot Heldout 90%* | Evaluation | Held-out clusters of sequences based on clustering at 90% sequence identity | 5,243 | 942 |
| *Swissprot Heldout 70%* | Evaluation | Held-out clusters of sequences based on clustering at 70% sequence identity | 5,428 | 818 |
| *Swissprot Heldout ECs* | Evaluation | Sequences from Swissprot that correspond to random held-out ECs, corresponding to those in the medium split for task 2 in the CARE benchmark (Yang et al., 2024b) | 5,714 | 177 |
| *Swissprot Train* | Training | Everything remaining in Swissprot that is not held-out above | 152,763 | 4201 |
| *Train Common ECs* | Generation Evaluation | Randomly sampled ECs from the pool of ECs that correspond to >500 sequences in *Swissprot Train* | n/a | 24 |
| *Train Rare ECs* | Generation Evaluation | Randomly sampled ECs from the pool of ECs that correspond to <10 sequences in *Swissprot Train* | n/a | 24 |
| *Heldout ECs* | Generation Evaluation | Randomly sampled ECs from the pool of ECs in *Swissprot Heldout ECs* | n/a | 24 |

The training objective is autoregressive residue prediction, and model training is flexible to any type of representation as conditioning. During inference, sequences are generated conditioned on a target function, which is represented in conditioning space (Fig. 1B). In this study, we explore conditioning using (1) enzyme function associated with the EC number, (2) taxonomy, and (3) textual descriptions of protein function. Multiple conditions can be considered jointly (Fig. A.2). More details are provided in Section A.2.

ProCALM aims to address the limitations of existing approaches for conditional sequence generation (Table 1). Unlike models trained from scratch with prompts, using *Conditional Adapters* is advantageous as it transfers knowledge from the pretrained ProGen2 model. This approach may also enable better OOD generalization compared to *Prompting* approaches, as the conditioning lies on a continuous conditioning space (Fig. 1C). Overall, ProCALM is a promising and flexible approach to unlock conditional generation of proteins with useful and unseen combinations of functions. ProCALM will be made publicly available.

## 4 RESULTS

To evaluate the ProCALM model and the quality of generated sequences, we built several train-test splits and selected several categories of ECs to generate from as conditioning, which are summarized in Table 2 and Fig. A.1. We focus our evaluation to three types of metrics:

1. **Generation quality.** Evaluated by measuring the fraction of generated sequences that look like valid enzymes based on sequence similarity to reference enzymes (*Valid Enzymes*) and the fraction of valid enzymes that have the *Correct* desired condition (e.g. EC number, taxonomy, etc.). Details of how enzyme validity and correct EC/taxonomy/function were determined are provided in Section A.3. *Enrichment* is a related metric, which describes

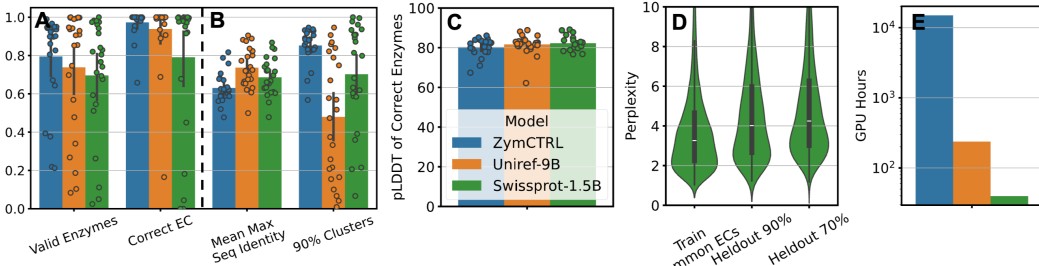

Figure 2: **ProCALM matches existing methods at generating sequences from common EC classes, at a lower cost.** **(A)** The generation quality of ProCALM is similar to existing methods (i.e., ZymCTRL), as measured by the fraction of generated sequences that resemble a known enzyme (*Valid Enzymes*) and by the fraction of those enzymes with the correct EC conditioning (*Correct EC*). **(B)** Generated, valid enzyme sequences have good diversity, as measured by the maximum sequence identity to a reference database (*Mean Max Seq Identity*) and the fraction of unique clusters (amongst themselves) at 90% sequence identity (*90% clusters*). **(C)** pLDDT, averaged across residues, of ESMFold-predicted structures is high and comparable across different methods. **(D)** The perplexities of protein sequences from common EC classes in the train set are similar to those held-out at 90% and 70% identity, which suggests robustness to overfitting. Comparatively, in ZymCTRL, the held-out sequences had much higher perplexities (Munsamy et al., 2024). **(E)** Number of GPU hours required to train ProCALM models is significantly lower than ZymCTRL. Swissprot-1.5B and Uniref-9B are ProCALM models trained on the *Swissprot Train* and *Uniref* datasets to 1.5 and 9 billion tokens, respectively, using the OH encoding of the EC number as a conditioning representation. Shorter training on the Swissprot dataset seems sufficient to achieve good performance. Error bars indicate standard deviation among 24 *Train Common ECs*.

the ratio between the fraction of generated sequences with a desired condition, compared to the prevalence of that condition in the train set. pLDDT averaged across all residues was also evaluated, to measure the confidence associated with structure prediction from ESMFold of generated sequences with correct conditioning. Other filters could also be used here (Johnson et al., 2024; Alamdari et al., 2023; Wang et al., 2024a; Nicolini et al., 2024).

2. **Generation diversity and novelty.** Evaluated by measuring the average maximum sequence identity (*Mean Max Seq ID*) to a reference database of sequences (high sequence identity suggests low novelty) and by the fraction of unique *Clusters* at 90% sequence identity (higher fraction indicates better diversity). The fraction of clusters is measured by clustering the generated, valid proteins among themselves.

3. **Robustness to overfitting.** Evaluated by measuring model *Perplexity* on sequences from different splits.

## 4.1 PROCALM MATCHES THE PERFORMANCE OF EXISTING METHODS

We first sought to understand if our approach could achieve similar performance to ZymCTRL, a state-of-the-art language model for generating enzyme sequences conditioned on EC number (Munsamy et al., 2024). We trained two different ProCALM models – one on 9 billion tokens of enzyme sequences from *Uniref*, and one on 1.5 billion tokens of enzyme sequences from *Swissprot Train*, with EC number represented as a one-hot (OH) encoding. While the model trained on Uniref for longer was better able to match the performance of ZymCTRL, we found that shorter training on Swissprot enabled sufficient generation quality (valid and correctly conditioned enzymes, Fig. 2A) and sequence generation with greater diversity (Fig. 2B). In general, training for longer increased the fraction of sequences with correct conditioning, but reduced sequence diversity (Fig. A.4). Because Swissprot has the highest quality annotations, we decided to do remaining analysis with models trained using the *Swissprot Train* dataset.

Parameter efficient finetuning is advantageous due to its low computational cost and greater robustness to overfitting (Sledzieski et al., 2024), compared to full parameter training. Impressively, our *Uniref-9B* and *Swissprot-1.5B* models only required 240 and 40 A100-hours to train, respectively, whereas ZymCTRL required 15,000 A100-hours (Fig. 2D). By comparison, ZymCTRL trained all model parameters for approximately 50 billion tokens and required a higher memory footprint than our

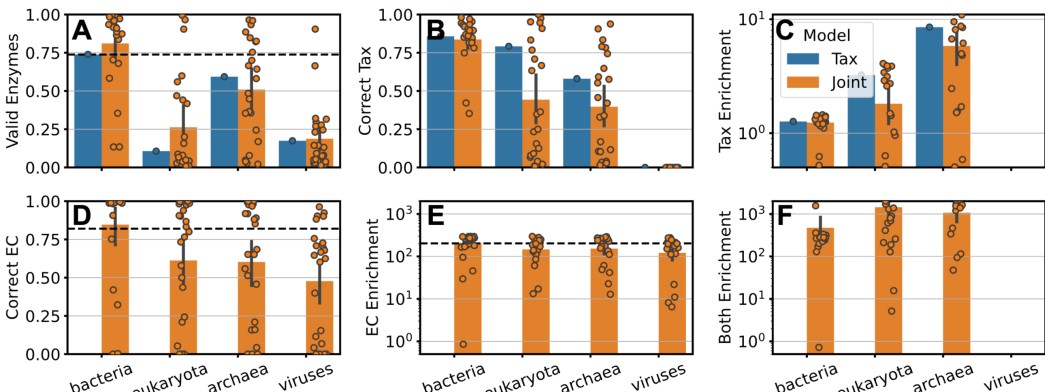

Figure 3: **ProCALM flexibly accommodates other types of conditioning, such as taxonomy (*Tax*) and joint EC-taxonomy (*Joint*) conditioning.** The *Tax* model was trained in the standard manner while the *Joint* model was trained with an architecture using parallel adapters (Fig. A.2). Taxonomy-conditioned and joint-conditioned models generate sequences with a high fraction of **(A)** valid enzymes, **(B)** correct taxonomy, and **(D)** correct EC. Generation from these models is also significantly enriched in those with the **(C)** correct taxonomy, **(E)** correct EC number, or **(F)** both, where enrichment is measured by the ratio of the prevalence among generated sequences to the prevalence in the training set. Some performance is sacrificed when learning to condition on multiple distributions simultaneously. EC and taxonomy were both represented using OH encoding, and models were trained to 9 billion tokens. Dashed line shows the performance of a model trained on EC conditioning only. Error bars indicate standard deviation among 24 *Train Common ECs*.

parameter-efficient training. Additionally, in ProCALM, held-out sequences demonstrate similar perplexities to the sequences associated with the most common ECs in the train set, which suggests that our model is not overfitting (Fig. 2C). By contrast, in ZymCTRL and our own experiments where we trained all model weights (Fig. A.3), the held-out sequences had significantly higher perplexity than common sequences in the train set. Finally, we examined all of these effects when scaling from ProGen2-Base (764 million parameters) to ProGen2-Large and ProGen2-XLarge, which have 2.7 and 6.4 billion parameters, respectively. Overall, we found that scaling to larger models resulted in lower overall losses (Fig. A.5) but did not significantly affect generation quality and diversity (Fig. A.6).

## 4.2 PROCALM ACCOMMODATES MULTIPLE TYPES OF CONDITIONING

ProCALM can use other types of information, such as taxonomy, to condition sequence generation. We trained two additional models, one conditioned on taxonomy, and one conditioned jointly on taxonomy and EC. For the jointly conditioned model, we used a modified architecture, where parallel adapters accept multiple sources of conditioning information (Fig. A.2). In an ablation study, we found that parallel adapters work better than merging the joint conditioning in the conditioning encoder and using a shared adapter (Fig. A.8). Overall, the taxonomy conditioned and jointly conditioned models enriched generation for the desired condition(s) (Fig. 3), showing that ProCALM is easily adaptable. Performance is highest for bacterial sequences, as they constitute the majority of the training data (Fig. A.1). However, learning multiple (EC and taxonomy) conditions sacrifices performance slightly compared to models trained to learn either EC or taxonomy conditioning alone.

To highlight the broad utility and inherent flexibility of ProCALM, we explored another task, conditional generation based on natural language prompts. While there are several models that enable text-guided protein sequence generation (Liu et al., 2024; Yin et al., 2024; Hayes et al., 2024; Praljak et al., 2024; Wang et al., 2024b), we focused on comparing to ProteinDT, which is publicly available and focuses on using natural language as conditioning. In ProteinDT (Liu et al., 2024), a multimodal representation is first learned between textual description and protein sequence using CLIP loss (ProteinCLAP representation), and this is decoded into a generated sequence using a PLM that has been subjected to indirect prompt tuning (finetuned to condition generation on different first token embeddings, i.e. conditions).

For fair comparison, we used the ProteinCLAP representation and the same training data as ProteinDT (proteins in Swissprot, not just enzymes) within our ProCALM framework to finetune ProGen2

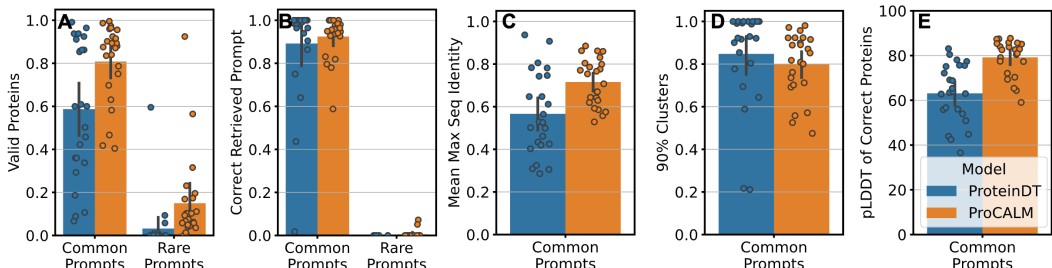

Figure 4: **ProCALM is flexible for natural language-guided generation of proteins with desired functions.** *Common Prompts* and *Rare Prompts* each refer to 24 randomly selected textual descriptions that have >500 instances and <10 instances, respectively, in the training set (Table A.1). In general, ProCALM generates sequences with higher rates of **(A)** valid protein sequences and **(B)** valid sequences mapping to the correct function (textual description), compared to ProteinDT. For ProCALM-generated valid proteins, their **(C)** novelty is a bit lower than ProteinDT but their **(D)** diversity is similar. **(E)** The pLDDT of predicted structures (via ESMFold) of generated sequences is higher for ProCALM. For sequence generation from ProteinDT, we used the default model with autoregressive PLM decoding, which is trained via indirect prompt tuning.

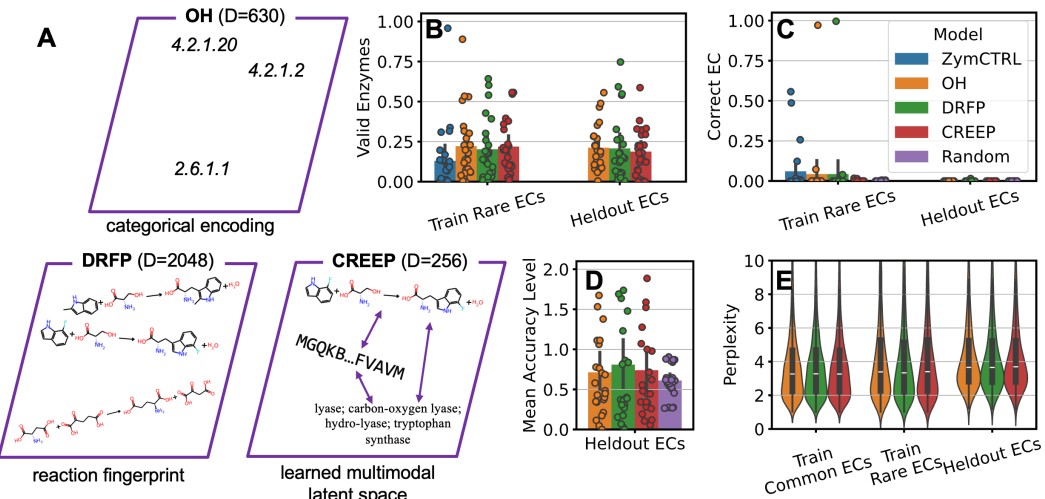

Figure 5: **ProCALM can perform out-of-domain generalization for rare and unseen EC classes.** **(A)** Three different representations of enzyme function explored in this work. OH encodes the hierarchy of EC numbers as a one-hot encoding, DRFP encodes a fingerprint of reactions associated to an EC number, and CREEP is a learned representation that encodes multiple modalities related to the EC number. **(B)** Our models generate a non-negligible fraction of *Valid Enzymes* for rare ECs in the train set and ECs that are entirely held out from the train set. **(C)** While almost none of the generated sequences using held-out ECs as conditioning mapped exactly to the target conditioning, **(D)** the generated sequences look more similar to the target EC. Accuracy level refers to the correctness of an EC. For example, if the target was 1.1.1.1, an EC of 1.2.1.2 would have an accuracy level of 1 and an EC of 1.1.1.2 would have an accuracy level of 3. **(E)** Perplexities are consistent for protein sequences from various splits, which suggests that the models are robust to overfitting. Models were trained to 1.5 billion tokens. Error bars indicate standard deviation among 24 ECs.

with adapters (Fig. 4). We then compared the generated sequences between both models, when generation was conditioned on *Common Prompts* and *Rare Prompts*, textual descriptions with many and few examples in the training set, respectively. ProCALM performed better than ProteinDT: it generated more valid and correctly conditioned sequences, without sacrificing the diversity of those sequences. Notably for *Rare Prompts*, while ProteinDT generates barely any valid proteins, ProCALM generalizes better and is still able to generate a decent fraction of valid proteins, though few are completely correctly conditioned.

### 4.3 PROCALM GENERALIZES TO OUT-OF-DISTRIBUTION CONDITIONS

We further explored ProCALM's ability to generalize, motivated by the fact that generative models able to generate sequences for novel, unseen functions would be highly impactful. For example, enzyme engineering often involves finding enzymes that can perform interesting non-native activities (Yang et al., 2024a;b; Chen & Arnold, 2020). We explored ProCALM's ability to generate sequences conditioned on rare and held-out EC numbers; the latter task has not been explored before. This setup reflects a real-world scenario where certain enzyme functions have not been annotated, but their corresponding genes have been sequenced. Thus, we explored ProCALM's ability to generalize and its flexibility to accommodate different representations of enzyme function. Namely, we encoded the EC number as a OH encoding of the EC hierarchy, a continuous fingerprint representation of reactions associated with an EC number (DRFP) (Probst et al., 2022), and a multimodal embedding from contrastive learning (CREEP) (Yang et al., 2024b) – visualized in Fig. 5A with more details in Section A.1.

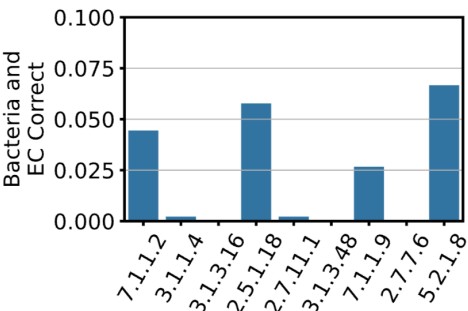

Overall, ProCALM can generate a substantial fraction of valid enzymes under this challenging conditioning (Fig. 5B). While a negligible number of generated sequences mapped to the correct EC number completely, the functions associated with the generated sequences were generally more similar to the target function (Fig. 5C-D). Using more continuous (DRFP and CREEP) representations of enzyme function did not result in better generalization performance. Held-out EC numbers with more similar functions to the training set generally yielded sequences with more similar functions to the target conditioning (Fig. A.9), and in general, the CREEP representation seems to capture somewhat different information than OH and DRFP. We note that there is no data leakage in the CREEP representation, as it was trained with the same held-out EC numbers. Finally, the perplexities of sequences associated with rare and held-out ECs are comparable to the train set, suggesting that our models are not prone to overfitting (Fig. 5E). Still, there is signifcant room for future research and improvement here.

Figure 6: **ProCALM has the potential to generate sequences with unseen taxonomy for certain EC classes.** The fraction of sequences successfully generated with the correct EC number and mapping to bacteria, when prompted with bacteria as conditioning. EC and taxonomy were both represented using OH encoding, and the model was trained to 1.5 billion tokens using parallel adapters with bacterial sequences from these EC classes held-out. ECs shown were selected based on the most common EC classes after having bacterial sequences removed.

Additionally, generating unseen combinations of taxonomy and enzymatic activity would also be interesting for applications such as discovering gene-editing enzymes in new organisms (Burstein et al., 2016). Thus, we next used ProCALM to generate bacterial sequences from several common EC classes, where all bacterial sequences in those EC numbers were held-out during model training. This scenario reflects a hypothetical real-world situation where a eukaryotic enzyme has not been discovered and annotated in a prokaryote. For certain EC classes, ProCALM could generate bacterial sequences (Fig. 6). Generation quality is better when the model is prompted with bacteria as conditioning (as opposed to other taxa), suggesting that the correct generations are not due to chance (Fig. A.10). Overall, ProCALM seems to be able to learn the distribution of bacterial sequences, allowing for some level of OOD generalization.

## 5 DISCUSSION

We have shown that conditional adapters are an efficient, flexible, and generalizable approach for enabling language models to perform steerable sequence generation. Our specific implementation, ProCALM, demonstrates strong performance for enzyme sequence generation conditioned on known EC numbers, at a significantly lower computational cost than prior approaches. ProCALM easily adapts to different types of conditioning (beyond protein structure), such as taxonomy and textual description of function, and to different representations for the same condition, such as OH, DRFP,

and CREEP representations of enzyme function. Our approach is also parameter efficient, resulting in lower memory usage and potentially faster training.

Most usefully, ProCALM can perform previously unexplored generation tasks, such as generating from unseen EC classes. Still, performance is limited, and there is significant room for improvement in this area. For certain EC classes, ProCALM was able to generate bacterial sequences corresponding to the correct EC, despite not having bacterial sequences belonging to those EC classes in the train set. 7.1.1.2 and 7.1.1.9 are transmembrane enzymes part of large complexes, but it is not clear why these particular functions were successfully generated. In the future, a more in depth examination of success and failure modes for ProCALM will be interesting.

For known functions, it is relatively straightforward to use function prediction tools such as BLAST to filter generated sequences to the valid ones with correctly conditioned function, though future work will benefit from better function prediction oracles for evaluation (Yu et al., 2023; Ayres et al., 2024; Huo et al., 2024). Future evaluation may also include more detailed analysis of generation quality (Ye et al., 2024) using various tools, with wet-lab validation. In particular, evaluation would benefit from a deeper consideration of how to measure similarity between functions such as EC classes or textual descriptions.

In general, we believe that the most interesting research direction for future improvement is generation of sequences with OOD functions and/or properties. In real-world application, evaluation is particularly intractable for these unannotated functions, so it will be even more critical to increase the hit rate for enzyme discovery on those functions. Having a smooth and navigable latent space for OOD functions will be especially important, along with training data encompassing more comprehensive annotations of natural and engineered proteins. While we explored various representations here based on fingerprints and contrastive learning, further exploration of multiple modalities such as structure, sequence, function, and natural language with different representation learning techniques could lead to breakthroughs (Yang et al., 2024b; Mikhael et al., 2024; Ramesh et al., 2022; Yin et al., 2024). For convenience, we finetuned our model using annotated proteins in Swissprot, but greater sequence diversity by incorporating sequences from unannotated and metagenomic datasets could be beneficial in the future.

Overall, we have demonstrated that using conditional adapters to finetune language models offers several advantages, including computational efficiency, flexibility to many types of conditioning, and generalization for OOD generation. This approach could easily be applied to other autoregressive PLMs (Hesslow et al., 2022; Ferruz et al., 2022). ProCALM will be publicly available, and we encourage researchers to apply our approach to other language models and further explore its potential for conditional protein generation.

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

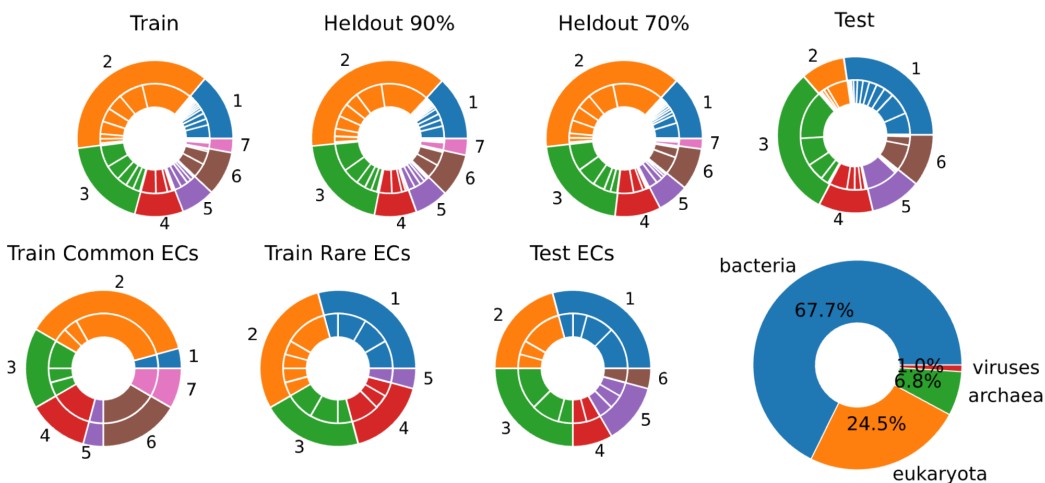

Figure A.1: Distributions of the datasets and splits used for ProCALM training and evaluation, broken down by EC class. Top row shows the distribution of protein sequences while bottom row shows the distribution of 24 EC numbers in each pie chart. Test refers to the held-out ECs. Taxonomy breakdown applies to the train set.

# A APPENDIX

## A.1 DATASETS AND REPRESENTATIONS

Uniref and Swissprot were downloaded from Uniprot (The UniProt Consortium et al., 2023) as all sequences with EC numbers on June 17, 2024. We left Uniref unprocessed for training. For Swissprot, we only considered the sequences corresponding to EC numbers with associated reactions from CARE (Yang et al., 2024b), and we first resampled it by weighting each sample with $\frac{1}{1+\ln(n)}$ relative probability of being sampled, where $n$ is the size of the cluster that a sequence belongs to when clustered at 50% identity using MMseqs2 (Steinegger & Söding, 2017). Effectively, this up-samples the sequences from smaller clusters. Afterward, we generated the held-out splits outlined in Table 2 and Fig. A.1. The remaining non held-out sequences in *Swissprot Train* were then used for training most models. For Fig. 6, we updated *Swissprot Train* slightly by also holding out bacterial sequences associated with the 9 EC classes explored.

Different representations of EC numbers were obtained for the Swissprot dataset. For OH, each of the four levels of the EC number were OH-encoded and concatenated, resulting in a 630-dimensional vector. The EC number is a hierarchical scheme consisting of four levels where each subsequent level describes enzyme function more specifically. For DRFP, DRFP representations (Probst et al., 2022) with dimension 2048 were obtained for all reactions associated to each EC based on CARE (Yang et al., 2024b), and the EC was encoded as the mean DRFP representation. DRFP is a fingerprint associated with the set change from reactants to products and has shown state-of-the-art ability for reaction classification. For CREEP, we used the model trained with default parameters on the medium train-test split for Task 2, which was trained using contrastive learning to align representations of protein sequences, reactions, and textual description (Yang et al., 2024b). We note that there is no data leakage in the CREEP representation, as it was trained with the same held-out EC numbers. From this model, we extracted representations for the textual description associated to each EC number.

The OH representation of taxonomy was simply encoded as a four-dimensional vector to describe the four possible kingdoms: bacteria, eukaryota, archaea, and viruses. While we also explored more specific encodings of taxonomic hierarchy, generation quality was not as good, as this conditioning information was more difficult to learn.

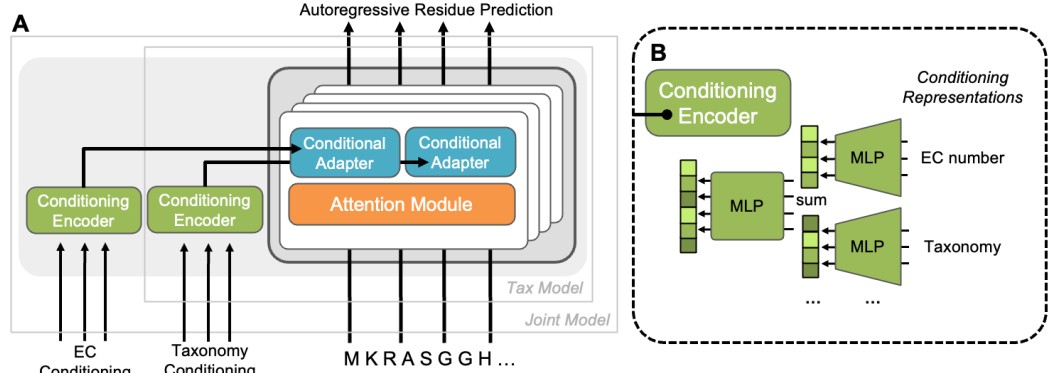

Figure A.2: Modification of ProCALM for joint conditioning using **(A)** parallel adapter modules or **(B)** a shared adapter module. For shared adapters, the conditioning information from multiple sources is merged in the conditioning encoder using non-linear transforms. In the parallel adapter module, conditioning information is instead passed through separate adapter layers and only merged at the end of each transformer layer via summation.

## A.2 ARCHITECTURE AND TRAINING DETAILS

Unless otherwise noted, we finetuned ProGen2-Base, which is a decoder-only autoregressive language model consisting of 27 layers, an embedding dimension of 1536, and 764 million parameters (Nijkamp et al., 2023). By default, the conditioning encoder consisted of two layers with a dimension of 256 each. The output was then concatenated with a low-rank projection of the language hidden state (dimension 16). Each adapter MLP consisted of 3 layers with dimension 2*(16 + 256) and a final layer with dimension (256 + 16). For the default ProCALM model, the adapter layers and conditioning encoder together consisted of approximately 70 million additional parameters, depending on the condition dimension. The adapter parameters dominated compared to the conditioning encoder weights. During parameter efficient training, the original ProGen2 weights were fixed, and the adapter and conditioning encoder were trained (pseudocode provided by Ruffolo et al. (2024a)). During full finetuning, all weights were trained. For the *Small* model, dimensions were reduced by half and only two layers were used in the adapter MLP. For the parallel adapter architecture, conditioning information was passed through separate adapters before being summed as the embedding update (Fig. A.2), resulting in approximately twice the number of trainable parameters. We also briefly explored the impact of scaling to larger ProGen2 models.

We used composer to perform training across 4 40GB A100s using distributed data parallel, and batches were sampled to minimize the usage of padding tokens. Each batch consisted of 144k tokens. Training took about 10 hours for every 1.5 billion tokens. One epoch for the processed *Swissprot Train* dataset corresponds to about $6 \times 10^7$ tokens.

## A.3 EVALUATION METRICS

The conditions (EC numbers and natural language prompts) used for generation evaluation are given in Table **??**.

Sequences were generated with probabilistic decoding with a top-p value of 0.95 and a temperature of 0.3, except for the text-guided generation task, which used a temperature of 1.0. To determine the validity and function of generated sequences, we used Diamond BLASTp (Buchfink et al., 2021). Our reference database consisted of 550k proteins from Swissprot, not just enzymes. For the main text figures, 900 sequences were generated for each unique condition, and for the remaining figures (including the text-guided conditioning task), 225 sequences were generated for each condition to get good statistics. *Valid* proteins were defined as those mapping to a hit in the reference database with default parameters, with at least 80% alignment coverage. A valid protein was determined to have a *Correct* EC or function if its BLASTp hit in the reference database mapped to the target EC number or natural language prompt.

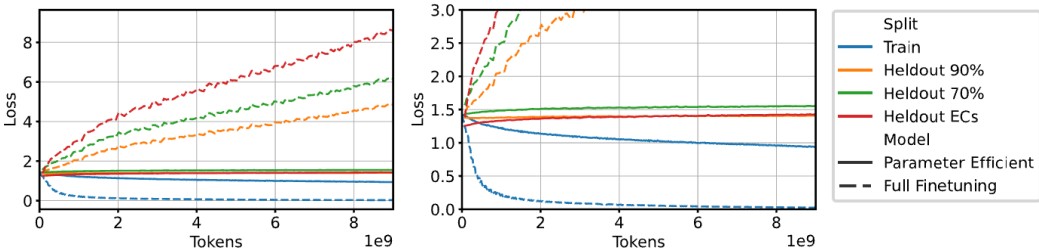

Figure A.3: Full parameter finetuning is prone to overfitting, based on the training loss curves. For the parameter efficient models, losses are stable on the held-out splits. For the fully finetuned models, losses increase rapidly on the held-out splits. Plots are shown for the model trained using OH encodings of EC numbers as conditioning on the *Swissprot Train* dataset. Training loss is measured as cross-entropy loss.

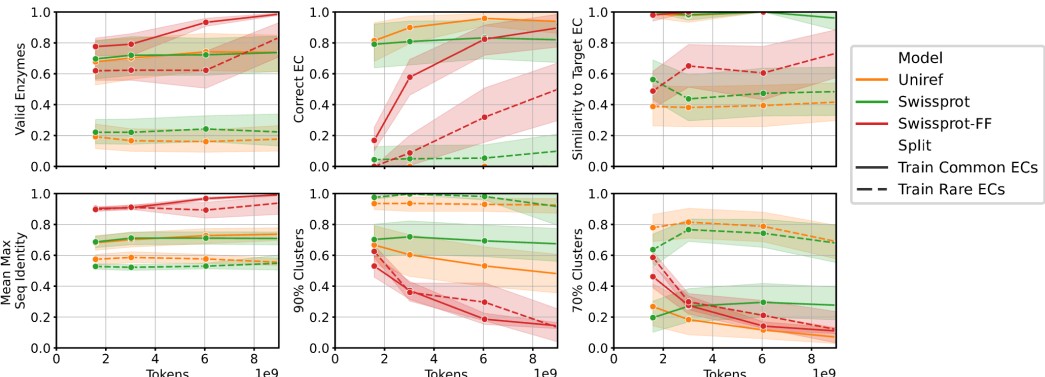

Figure A.4: The parameter efficient models are generally more robust to overfitting, as measured by the quality and diversity of sequences generated at different stages of training. In general, as training progresses, the fraction generated with correct conditioning goes up, but the diversity goes down. Uniref and Swissprot refer to the parameter efficient models trained using OH encodings of EC numbers as conditioning on the Uniref and Swissprot datasets, respectively. Swissprot-FF refers to the model where all parameters were finetuned on Swissprot. Performance is measured for sequences generated conditioned on *Train Common ECs* and *Train Rare ECs*. Error bars indicate standard deviation among 24 ECs.

*Mean Max Seq ID* was defined as the average maximum sequence identity of generated sequences, compared to the reference set of 550k proteins. The fraction of *X% Clusters* was measured by clustering the generated sequences at X% sequence identity using MMseqs2 (Steinegger & Söding, 2017). We also used MMseqs2 as a taxonomy oracle (Mirdita et al., 2021), which assigns taxonomy based on the lowest common ancestor inferred by placing a query sequence in a phylogenetic tree using Swissprot as a reference database. We note that this may not be the most effective oracle for generated sequences, as a non-negligible fraction of query sequences could not be assigned taxonomy.

## A.4 ADDITIONAL RESULTS

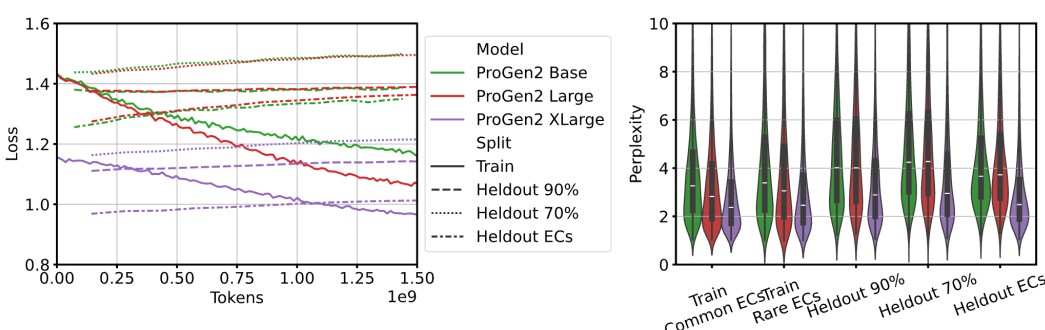

Figure A.5: Losses and perplexities of ProCALM models, for various model sizes and training and test splits. Overall, the ProCALM model based on ProGen2 XLarge shows lower losses and perplexities. All models were trained using *Swissprot Train* and default parameters. Perplexity was measured on the models trained to 1.5 billion tokens.

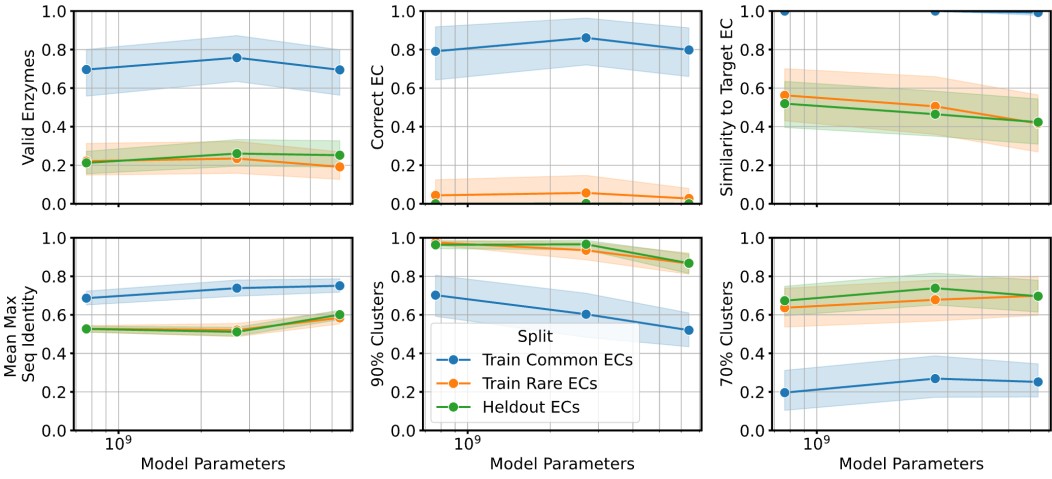

Figure A.6: Various metrics used to evaluate the quality, diversity, and novelty of generated sequences, as a function of model size. 3 different pretrained models were used: ProGen2-Base (764 million parameters), ProGen2-Large (2.7 billion parameters), and ProGen2-XLarge (6.4 billion parameters). Overall, scaling does not seem to have significant effects on the quality of generated sequences.

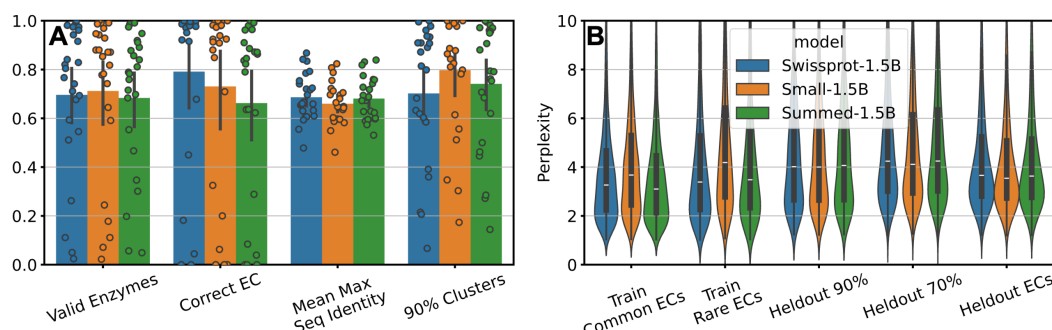

Figure A.7: Ablation study showing certain design choices for the ProCALM architecture. **(A)** The generation quality of various architectures as measured by the fraction of generated sequences that are (*Valid Enzymes*) and by the fraction of those enzymes with the correct EC conditioning (*Correct EC*). The diversity of generated sequences as measured by maximum sequence identity to a reference database (*Mean Max Seq Identity*) and the fraction of unique clusters at 90% sequence identity (*90% Clusters*). Error bars indicate standard deviation among 24 Train Common ECs. **(B)** The perplexities of protein sequences from common EC classes in the train set, compared to those from rare ECs, those held-out at 90% and 70% identity, and those with entirely held-out EC numbers. The standard Swissprot model is compared to a smaller architecture with a quarter of the adapter parameters (*Small*) and a model where the low-rank adapter embedding is summed with the low-rank language model embedding, instead of concatenation (*Summed*). All models were trained using EC conditioning with OH encoding to 1.5 billion tokens.

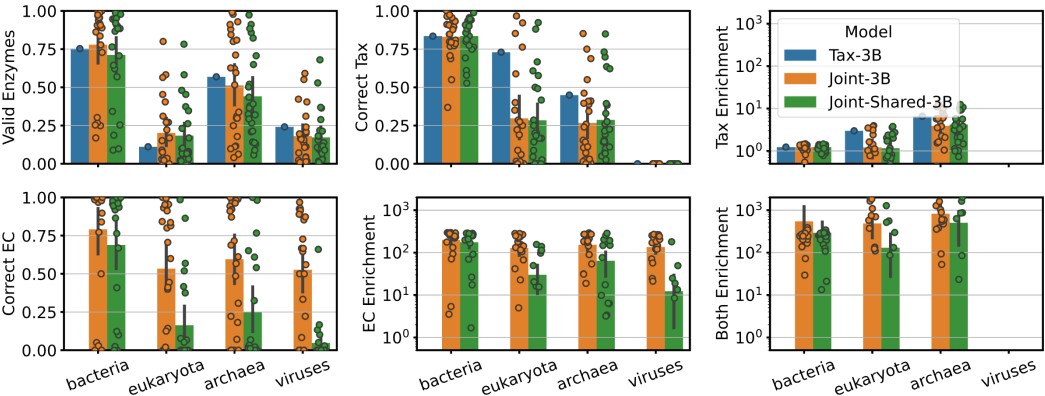

Figure A.8: Ablation study showing design choices for ProCALM architecture for taxonomy and EC conditioning. A *Joint-Shared* adapter is compared to using parallel adapters (*Joint*). Performance is measured by the fraction of generated sequences that correspond to valid enzymes, correct taxonomy, and correct EC. For the jointly conditioned models, using parallel adapters (orange) results in better in-distribution generation quality compared to using a shared adapter architecture (green). The *Tax* model was trained with taxonomy conditioning only. All models were trained to 3 billion tokens using OH encoding of the conditions. Error bars indicate standard deviation across 24 Train Common ECs.

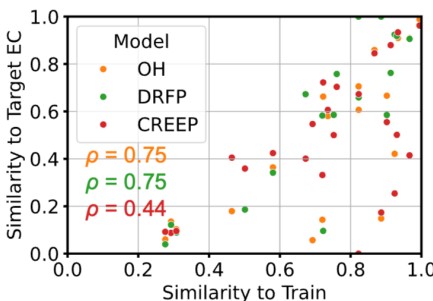

Figure A.9: The maximum similarity of a target EC to ECs in the train set (x-axis) is generally correlated to mean similarity of ECs of generated sequences to the target EC (y-axis). Similarity is measured using cosine similarity and mean DRFP encoding of the EC number.

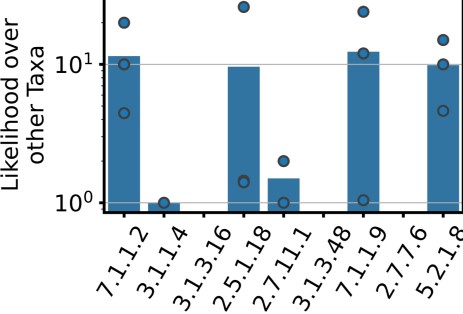

Figure A.10: *Likelihood over other Taxa* describes the ratio of the fraction of generated sequences with the correct conditioning when being prompted with bacteria vs when being prompted with other taxa, where correct conditioning is the fraction of sequences with the correct EC and mapping to bacteria (Fig. 6). A ratio greater than 1 suggests that the correctly generated sequences are not being found by chance. Error bars indicate standard deviation among 3 other taxa.

Table A.1: Example prompts used to evaluate natural language-conditioned generation from PLMs.

**Common Prompts**

carbamoyl phosphate + L-aspartate = H(+) + N-carbamoyl-L-aspartate + phosphate Pyrimidine metabolism; UMP biosynthesis via de novo pathway; (S)-dihydroorotate from bicarbonate: step 2/3. Belongs to the aspartate/ornithine carbamoyltransferase superfamily. ATCase family.

ATP + L-cysteine + tRNA(Cys) = AMP + diphosphate + L-cysteinyl-tRNA(Cys) Binds 1 zinc ion per subunit. Monomer. Belongs to the class-I aminoacyl-tRNA synthetase family.

ATP-dependent specificity component of the Clp protease. It directs the protease to specific substrates. Can perform chaperone functions in the absence of ClpP. Component of the ClpX-ClpP complex. Forms a hexameric ring that, in the presence of ATP, binds to fourteen ClpP subunits assembled into a disk-like structure with a central cavity, resembling the structure of eukaryotic proteasomes. Belongs to the ClpX chaperone family.

Involved in protein export. Acts as a chaperone by maintaining the newly synthesized protein in an open conformation. Functions as a peptidyl-prolyl cis-trans isomerase. [protein]-peptidylproline (omega=180) = [protein]-peptidylproline (omega=0) About half TF is bound to the ribosome near the polypeptide exit tunnel while the other half is free in the cytoplasm. Consists of 3 domains; the N-terminus binds the ribosome, the middle domain has PPIase activity, while the C-terminus has intrinsic chaperone activity on its own. Belongs to the FKBP-type PPIase family. Tig subfamily.

Binds directly to 16S ribosomal RNA. Belongs to the bacterial ribosomal protein bS20 family.

ATP + L-histidine + tRNA(His) = AMP + diphosphate + H(+) + L-histidyl-tRNA(His) Homodimer. Belongs to the class-II aminoacyl-tRNA synthetase family.

**Rare Prompts**

Involved in the regulation of gene expression by abscisic acid, stress factors and by components of stress signal transduction pathways. Transcription factor that binds to the GCC-box pathogenesis-related promoter element. Part of a transcriptional repressor complex including a histone deacetylase. Interacts with SIN3 and HDA19. Contains a slightly degenerated ERF-associated amphiphilic repression (EAR) motif, which may be involved in the activity of transcriptional repression. Phosphorylated by PKS3. Belongs to the AP2/ERF transcription factor family. ERF subfamily.

Functions as a component of the Five Friends of Methylated CHTOP (5FMC) complex; the 5FMC complex is recruited to ZNF148 by methylated CHTOP, leading to desumoylation of ZNF148 and subsequent transactivation of ZNF148 target genes (By similarity). Component of the PELP1 complex involved in the nucleolar steps of 28S rRNA maturation and the subsequent nucleoplasmic transit of the pre-60S ribosomal subunit (By similarity). May play a role during development (By similarity). Component of the 5FMC complex, at least composed of PELP1, LAS1L, TEX10, WDR18 and SENP3; the complex interacts with methylated CHTOP and ZNF148. Interacts with NOL9. Component of the PELP1 complex, composed of at least PELP1, TEX10 and WDR18. The complex interacts with pre-60S ribosome particles. Mainly found in the nucleoplasm, with low levels detected in the cytoplasmic and chromatin fractions. Belongs to the WD repeat IPI3/WDR18 family.

Involved in the final reduction of the elongation cycle of fatty acid synthesis (FAS II). Catalyzes the NADH-dependent reduction of a carbon-carbon double bond in an enoyl moiety that is covalently linked to an acyl carrier protein (ACP). It can use both crotonyl-CoA and crotonyl-ACP. a 2,3-saturated acyl-[ACP] + NAD(+) = a (2E)-enoyl-[ACP] + H(+) + NADH a 2,3-saturated acyl-CoA + NAD(+) = a (2E)-enoyl-CoA + H(+) + NADH (2E)-butenoyl-[ACP] + H(+) + NADH = butanoyl-[ACP] + NAD(+) butanoyl-CoA + NAD(+) = (2E)-butenoyl-CoA + H(+) + NADH Weakly inhibited by triclosan. Lipid metabolism; fatty acid biosynthesis. Monomer. Belongs to the TER reductase family.

Translation initiation regulator which represses repeat-associated non-AUG (RAN) initiated translation probably by acting as a competitive inhibitor of eukaryotic translation initiation factor 5 (EIF5) function (PubMed:29470543, PubMed:34260931). Enhances histone H4 gene transcription but does not seem to bind DNA directly (PubMed:11524015). Expressed in day 3 embryo. Belongs to the BZW family.

