# OpenReview forum: "Conditional Enzyme Generation Using Protein Language Models with Adapters"
_ICLR.cc/2025/Conference — Submitted to ICLR 2025_

### Official Review · Reviewer_E3g5 · 2024-10-30

**Soundness:** 3
**Presentation:** 3
**Contribution:** 3
**Rating:** 8
**Confidence:** 4

**Summary:**

The authors propose ProCALM, an adapter-based method to allow for conditional generation of proteins from pre-trained protein language models, and demonstrate the performance of ProCALM on conditional enzyme generation. Their main contributions are 1) demonstrating that conditional adapters, while being parameter efficient and inexpensive to train in both time and memory, yield comparable performance to existing methods and 2) creating a flexible framework for conditional generation -- ProCALM is not limited to any one particular class of condition, and the authors demonstrate individual and joint conditioning across enzyme class and taxonomy. Lastly, the authors assess the ability of ProCALM to generalize to rare and completely held-out/out-of-distribution enzymes (this includes unseen combinations of taxonomy and enzyme class). The authors also evaluate across a broad range of metrics and perform several ablation experiments.

**Strengths:**

1. The paper is written very clearly and does a good job of contextualizing the contributions of ProCALM against prior work.
2. While this paper does not represent the first use of conditional adapters for protein generation, the authors explore conditioning on information other than structure (like taxonomy and enzyme class)
3. The evaluation metrics are well-chosen and reasonable for sequence-level similarity and diversity, though it may be worthwhile to perform more fine-grained evaluation on generated sequences (i.e. domain-level)
4. Based off the evaluations provided, ProCALM is clearly performant and compares well to the existing method tested, ZymCTRL
5. The authors perform an ablation study to test the benefits of parallel adapters vs. a shared adapter in the case of conditioning both taxonomy and enzyme class,

**Weaknesses:**

My main concern is about the out-of-distribution (OOD) generalization claims, which I believe are currently unsupported by the evaluations/data-splitting:
1. The 70% and 90% sequence identity thresholds used for clustering to get held-out clusters for evaluation (Table 2) are really high thresholds.
- It seems like it would be totally feasible for a sequence in train and a sequence in one of these evaluation sets to be very closely related homologs (~40% shared sequence similarity and higher).
- In such a case, it seems like ProCALM could simply generate, for a given enzyme commission (EC) class, a sequence very similar to one seen with the same EC class in training.
- I believe that adding evaluations for held-out clusters at other thresholds will strengthen the overall results of this work.

2. The Heldout ECs for generation evaluation are randomly sampled ECs, which doesn't account for the hierarchy of EC numbers.
- While it would not be reasonable to hold out say all sequences where the first digit of the EC number starts with a 7, there should be some standardizing of what level of EC number gets held out. For example, maybe if all sub-subclasses are held out for a specific subclass, that would be an informative evaluation to see.
- In light of the random sampling of ECs for the heldout group, the mean accuracy level in Figure 4.D seems a little underwhelming. If the mean accuracy level is below 1, then the accuracy level is more about the ability of ProCALM to generate examples of the same overall class, which probably relies more on overall sequence similarity between examples of the target class and the training data, which comes back to the point about high percent identity clustering thresholds.

**Questions:**

I think that the authors make a convincing case that ProCALM is broadly applicable and useful given that it is 1) parameter efficient and inexpensive to train while matching ZymCTRL and 2) flexible to allow multiple types of conditioning. I also think that the paper would benefit from addressing the concerns about the OOD generalization claims.

I may have misunderstood the data-splitting, evaluations, or something else entirely, and I am willing to increase my score if this is the case and/or the authors address my concerns listed in the weaknesses section.

I'm also curious if the authors have examined the properties of the subsequences in the ProCALM-generated enzymes. For example, a histidine kinase will contain a histidine kinase domain (which may be as short as 70 amino acids), and this is probably the region of the protein to which it is most important to generate something similar in order to yield a working histidine kinase. I think it would be valuable to see if the generated sequences for a specific EC class contain the domain(s) that the target sequences contain and in what order (for multi-domain enzymes)

**Note:** raised rating to an 8 in light of authors' responses and revisions

---

> ### Author Response · Authors · 2024-11-13
>
> *Our Response to (1): Thank you for your support of our study and for the insightful feedback. We would first like to clarify our dataset splitting details. In Table 2, "Swissprot Heldout 70%" and "Swissprot Heldout 90%" are held-out protein sequences used only for the evaluation of perplexity. By contrast, we used the "Swissprot Heldout ECs" split to evaluate generation quality from OOD EC numbers. This split was constructed by holding out all protein sequences associated with certain EC numbers. These sequences are rarely homologous to the training sequences (as they belong to different enzyme families entirely), and nearly all have <40% identity to the training set. Therefore, when we ask the model to generate EC numbers that are OOD (not present in the training set), this is a very difficult task, and we see underwhelming performance in Figure 4D. We must admit that there is a lot of room for improvement here; by being one of the first to explore this question, we encourage others to tackle it with different approaches.*
>
> *Our Response to (2): As for the heldout ECs, we note that they are sampled across diverse EC numbers at level 4 (X.X.X.X), based on the "medium" split for Task 2 in the CARE benchmarks (https://arxiv.org/abs/2406.15669v1, details in Table 3 and Figure A.3B). In fact, these EC numbers are intentionally spread across functional space (details in CARE Benchmarks Appendix A.2). This essentially means that if 4.2.1.20 is held-out, there will still be sequences with other EC numbers falling under 4.2.1.- (similar functions) in the training set. You may also note that the "hard" split in the CARE benchmarks is more difficult because all EC numbers under a certain EC level 3 (X.X.X.-) are held out. Thus, the most similar EC numbers still present in the training set in that case would be X.X.-.- . However, this presents a more difficult task, so we only explored the "medium" split in this study. We hope this clarifies why there is no data leakage and that the held-out EC numbers were thoughtfully chosen to be spread across function and sequence space. Happy to explain further if there are remaining questions and/or concerns!*
>
> *The reviewer also raises an interesting point about more detailed analysis of the generated sequences. It is correct that certain enzymes have important domains, such as active sites, which may be more highly conserved. However, because most enzymes have complex and poorly understood sequence-function relationships, we have not had a chance to examine these subdomains in detail and primarily focused evaluation on local sequence alignment with BLAST. That being said, we are currently working on structure evaluation of generated sequences (using pLDDT), which we be included in this revision as another metric to evaluate generation quality. We are also performing additional experiments to strengthen the overall study, as suggested by the other reviewers.*
>
> *Please let us know if you have any further questions or concerns that we can address.*

---

> > ### Author Response · Authors · 2024-11-22
> > **New results and revised manuscript**
> >
> > Please additionally note the following additions (revision is uploaded as a new PDF file), which we believe help make our study more compelling:
> >
> > 1. To highlight the broad utility of ProCALM: We have trained new ProCALM models that utilize textual descriptions to generate proteins more broadly, not just enzymes. After benchmarking against ProteinDT, we found that ProCALM generalizes better and generates higher quality sequences **(see Figure 4)**.
> > 2. To provide additional insights on the ML side: We have explored scaling effects for ProCALM by training larger models and found that larger models do not necessarily result in improved performance **(see Figure A.5 and A.6)**.
> > 2. To make the evaluation more convincing: We have evaluated the generated sequences with pLDDT and found that our generated sequences have structures with high predicted confidence and comparable to or better than existing methods **(see Figure 2C and Figure 4E)**.

---

> > ### Comment · Reviewer_E3g5 · 2024-11-24
> >
> > Thank you for the clarification on my questions about the Swissprot Heldout 70% and 90% evaluation datasets as well as the CARE benchmarks!
> >
> > I still have some confusion about the heldout EC generation results however. Do you have any intuition for why the mean accuracy for generation is below 1? I realize that the model is tasked with generating EC numbers that are not present in the training set, but since the heldout ECs are sampled at level 4, I would expect that generated sequences would at least consistently be in the same enzyme class. This feels counterintuitive to the results provided in Fig. 2A and Fig. 3D, where ProCALM can generate sequences with the correct EC number fairly reliably. Even if the results for the common, not heldout EC numbers could be attributed partially due to memorization or leakage, why would ProCALM not generate a sequence from the same class as the desired EC number?

---

> > > ### Author Response · Authors · 2024-11-25
> > > **Explanation for low performance on heldout EC generation**
> > >
> > > Thanks for these additional questions. It is correct to conclude that performance is much better in Fig.2A compared to Fig. 5D, likely because of some memorization or overfitting to the training data. The overfitting is likely exacerbated because more “similar” EC classes do not necessarily correspond to more similar protein sequences. However, you may notice that for some of the EC numbers, the mean accuracy level approaches 2.0 (Fig. 5D), which is not present in the random baseline.
> > >
> > > Still, we elaborate why we might be observing overfitting effects. Firstly, EC numbers are human-defined reaction classes, so the DRFP and CREEP encodings do not capture the same notion of similarity as EC numbers. In other words, two very similar reactions (quantified by DRFP) can actually have very different EC numbers (this was also observed in the CARE benchmarks paper, Figure 2B). Secondly, in ProCALM, the onehot-encoded EC number used for conditioning is 630 dimensions in total, but only 7 of those dimensions are used to describe the first, broadest EC level (X.-.-.-). By contrast, 425 of those dimensions are used to describe the last EC level (-.-.-.X), as there are significantly more categories at this level. Therefore, the onehot representation is not ideal either, as it likely results in overfitting to the most specific EC level, reducing generalizability. Further work is needed to develop better representations for functional conditioning, and concurrently, better metrics to evaluate similarity between functions (reactivities).
> > >
> > > We hope that this provides more intuition on this matter! We will update the manuscript accordingly to explain this better.

---

> > > > ### Comment · Reviewer_E3g5 · 2024-11-26
> > > >
> > > > Thank you for your explanation. In light of the authors' clarifications on my questions, the additional results, and the revisions to the manuscript, I raise my score to an 8

---

### Official Review · Reviewer_UYRw · 2024-11-04

**Soundness:** 3
**Presentation:** 3
**Contribution:** 2
**Rating:** 5
**Confidence:** 4

**Summary:**

The paper proposes ProCALM, a method for the conditional generation of proteins using protein language models enhanced with conditional adapters. This approach aims to generate protein sequences tailored to meet specific types or molecular functions.
The authors conduct several experiments and demonstrate that, for seen enzyme families, ProCALM achieves comparable to previous methods like ZymCTRL. Additionally, it shows the ability to generalize to rare and novel enzyme families and taxonomies.

**Strengths:**

1. Adapter is indeed a parameter-efficient method that enables controllable generation at a relatively low cost and effectively mitigates overfitting.
2. The authors conducted extensive experiments to demonstrate the effectiveness of the proposed method, such as generating unseen families, and some necessary ablation studies like the performance in different function representations.

**Weaknesses:**

1. Compared to previous fine-tuning models designed to generate protein sequences with specific functions, the approach tries to learn the mapping of functional distribution, and indeed holds some potential to generate sequence with functions not encountered during training. However, it seems to rely on the quality of the functional representation, and there seems to be a risk of “data leakage” during the stage of function representation learning.
2. Although ProCALM has a lower training cost than previous methods, its ability to generate sequences with both seen and unseen protein families is limited. For seen functions, its performance is slightly inferior to existing methods, and for unseen families, its capabilities remain quite limited.

**Questions:**

I have listed my concerns and questions in 'Weaknesses'.

---

> ### Author Response · Authors · 2024-11-22
> **Response to weaknesses**
>
> Our Response to (1): Thank you for taking the time to review our paper. We have made some significant revisions to our manuscript and uploaded the revision as a new PDF file. We would first like to clarify that we ensured splitting would not have data leakage. Namely, the OOD EC numbers evaluated during generation were entirely held out. The representations we used also do not have data leakage. The Onehot encodings and DRFP embeddings are fixed representations, thus they are not learned. While the CREEP representation is learned using contrastive loss, we clarify in the main text that “there is no data leakage in the CREEP representation, as it was trained with the same held-out EC numbers.” More details on the CREEP training procedure can be found at https://arxiv.org/abs/2406.15669v1.
>
> Our Response to (2): This is a valid point, although we would like to note that we are one of the first people to explore the task of generation of OOD enzyme functions. An important contribution of our study is showing how difficult this task is, while also highlighting its importance to the community. We approached this task not knowing if OOD generalization is even possible, given publicly available data. We hope that future work will be able to tackle this question with different approaches.
>
> That being said, we would like to underscore one of the main contributions of our study: showing that adapters are a framework for conditional generation from PLMs that is flexible to many types of conditioning. By contrast, previous studies using adapters for PLMs have been limited to conditioning on structure. To make the broader utility of ProCALM more compelling, in this revision, we have trained new ProCALM models that utilize natural-language descriptions to generate proteins more broadly, not just enzymes. After benchmarking against ProteinDT, we found that ProCALM generates higher quality sequences and generalizes better **(see Figure 4)**.
>
> Additionally, we’ve now evaluated the generated sequences with pLDDT and found that our generated sequences have structures with high predicted confidence and are comparable to or better than existing methods **(see Figure 2C and Figure 4E)**. We hope that these revisions convey how our study is compelling and useful. Please let us know if you have any further questions or concerns that we can address.

---

> > ### Author Response · Authors · 2024-11-29
> >
> > Dear reviewer,
> >
> > As we enter the last few days of the discussion period, we are checking in to see if you have had a chance to look over our revised manuscript; we believe our new results help make our study stronger and compelling overall. Please let us know if we have addressed your concerns and if you have any further questions.
> >
> > Best,
> >
> > Authors

---

### Official Review · Reviewer_HNwz · 2024-11-05

**Soundness:** 3
**Presentation:** 3
**Contribution:** 2
**Rating:** 5
**Confidence:** 4

**Summary:**

This paper proposes a novel approach, ProCALM (Protein Conditionally Adapted Language Model), for the conditional generation of proteins using adapters in protein language models. They leverage a seperate encoder to capture information of given condition into latent representation. The conditional adapter layer, presented within each transformer layer, integrates this latent representation with the language model's embeddings to ensure conditional generation. Experimental results demonstrate that ProCALM can successfully generate protein sequences conditioning on enzyme function and taxonomy, and it shows significant generalization capabilities to unseen enzyme families and taxonomies.

**Strengths:**

1. The proposed method ProCALM is parameter-efficient and computationally inexpensive to finetune protein language models for conditional generation tasks.
2. ProCALM can handle multiple types of conditioning information, such as enzyme function and taxonomy, and different representations of the same condition.
3. The paper shows that ProCALM can successfully generalize to out-of-distribution conditions.

**Weaknesses:**

1. This paper does not cite LM-Design [1], which introduces a lightweight adapter into protein language models to perform structure-conditioned sequence generation and is very related to this work.
2. The method proposed in this paper is similar to LM-Design, and the experimental results, such as generalization to OOD distribution, have already been demonstrated by LM-Design. Therefore, I believe this paper lacks sufficient novelty.
3. The current results still lag behind existing methods (such as ZymCTRL) in terms of generation quality, diversity and perplexity, and only shows advantage in training costs, which may not be sufficiently strong to prove the effectiveness of this method.
4. The evaluation metrics are not comprehensive enough. For sequence quality, the pLDDT metric has widely been used in sequence generation evaluation [2,3]. And for sequence diversity, comparing generated sequences among themselves, rather than against a reference database, can more directly reflect diversity. For example, this can be evaluated from both sequence and structure dimensions: by calculating pair-wise sequence identity and by using a structure prediction model (such as ESMFold) to first obtain the structure of the sequence and then calculating the pair-wise structural similarity.

[1] Zaixiang Zheng, Yifan Deng, Dongyu Xue, Yi Zhou, Fei YE, and Quanquan Gu. Structure-informed language models are protein designers.

[2] Sarah Alamdari, Nitya Thakkar, Rianne van den Berg, Alex Xijie Lu, Nicolo Fusi, Ava Pardis Amini, and Kevin K Yang. Protein generation with evolutionary diffusion: sequence is all you need.

[3] Xinyou Wang, Zaixiang Zheng, Fei Ye, Dongyu Xue, Shujian Huang, and Quanquan Gu. Diffusion language models are versatile protein learners.

**Questions:**

1. Previous work [1] has validated the effectiveness of scaling. Considering that ProGen2 also has 2.7B and 6.4B versions, can scaling to bigger protein language model further improve the generation performance of the method proposed in this paper?

---

> ### Author Response · Authors · 2024-11-22
> **Response to weaknesses and questions**
>
> Our Response to (1) and (2): Thank you for the detailed feedback. We have made some significant revisions to our manuscript and uploaded the revision as a new PDF file. In our revision, we have included a more detailed discussion of previous works in the context of our current work, including a discussion and citation of LM-Design **(see Related Work)**. While previous studies using adapters have been largely limited to conditioning on protein structures (including LM-Design), we would like to highlight that ours (ProCALM) is the first study to explore flexible conditioning with multiple types of inputs and explore OOD generalization in function space. We have added the following text to the revision to better explain this: “LM-Design is particularly interesting because the authors also evaluate generalization toward unseen protein folds (Zheng et al., 2023). Conditioning on structure is useful, as structure often determines function, but a goal of protein engineering is often to find proteins with novel functions (such as for new-to-nature enzyme activity (Arnold, 2018)), without any known structure or sequence performing this function. Thus, there is a need to explore models that can condition directly on function and generalize to new functions.”
>
> Our Response to (3): To better showcase the broad utility of our approach, we have added a significant additional task with new experiments and evaluations. Namely, we trained a ProCALM model that can conditionally generate from natural-language descriptions of proteins and found that it performs better than ProteinDT, an existing model that can perform this task **(see Figure 4)**.
>
> Our Response to (4): We would like to clarify that we did in fact calculate the sequence diversity among themselves: this is given by the metric called "90% clusters" in Figure 2B, for example. The metric is calculated as (number of unique clusters at 90% identity clustering)/(number of total sequences). We have updated the text to explain this more clearly. Evaluating with pLDDT is a good suggestion, and we have performed additional analyses and reported those results in the revision, finding that our structures are as confident or more confident that existing approaches **(see Figure 2C and Figure 4E)** . Regarding structure diversity, since sequence space maps to structure space degenerately, most structures for a given enzyme family should be the same rather than different. We have now confirmed this by evaluating structure diversity with Foldseek (https://github.com/steineggerlab/foldseek) clustering. We found that, for a given enzyme class, typically the (number of unique structure clusters at 90% coverage)/(number of total structures) is less than 5%, which is consistent between ProCALM and ZymCTRL and not too surprising. Thus, we chose not to include these results in the manuscript.
>
> The suggestion to examine scaling effects is also interesting. We trained two new models with ProGen2 2.7B and 6.4B, but larger models did not necessarily result in improved performance. **Please see Figure A.5 and A.6.**
>
> We hope that these revisions convey how our study is compelling and useful, reiterating that we are the first to explore the use of adapters for conditioning PLMs beyond structure. Please let us know if you have any further questions or concerns that we can address.

---

> > ### Author Response · Authors · 2024-11-29
> >
> > Dear reviewer,
> >
> > As we enter the last few days of the discussion period, we are checking in to see if you have had a chance to look over our revised manuscript; we believe our new results help make our study stronger and compelling overall. Please let us know if we have addressed your concerns and if you have any further questions.
> >
> > Best,
> >
> > Authors

---

### Official Review · Reviewer_5p6F · 2024-11-08

**Soundness:** 2
**Presentation:** 2
**Contribution:** 2
**Rating:** 5
**Confidence:** 4

**Summary:**

The paper introduces ProCALM (Protein Conditionally Adapted Language Model), an approach for the conditional generation of proteins, specifically enzyme in this paper, using adapter tuning with protein language models.
In silico experiments show that matches existing methods (ie, ZymCTRL) in generating sequences from target enzyme families while offering several advantages, such as being parameter-efficient to train, allowing multiple conditionings (by adding more adapter and re-train accordingly(, and can generalize to rare and unseen enzyme families and taxonomies.
ProCALM demonstrates potential in generating sequences for unseen functions, although there is room for improvement and future research.

**Strengths:**

The use of adapters allows for parameter-efficient training, and easy accommodates various types of conditioning information, such as enzyme function and taxonomy for different applications.

ProCALM shows capabilities to generalize to rare and unseen enzyme families and taxonomies, which is an unique advantage over existing methods.

**Weaknesses:**

Despite the promise of ProCALM in generating functional protein sequences, the current form of the manuscript can be significantly improved if the following concerns are addressed in the future.

**Lack of technical novelty in the machine learning side:**
This paper appears to be a simple adaptation of adapter tuning for PLMs for one specific application scenario. Neither a new ML method nor new insights into how to properly tailor existing ML methods to protein problems are presented. Protein LMs (ProGen) are existing models, and adapter tuning is widely used in LMs, including protein LMs [1, 2, 3, 4]. The integration method seems fairly straightforward, with the only difference being the task (enzyme design).

**The performance is not strong and the evaluation is not sufficient**.
Compared to existing methods, ProCALM's performance is comparable and not impressive.
More efforts need to be put in providing clear evidence of ProCALM's superiority over existing methods.
Despite the promise in generating sequences for unseen functions, this paper only studies enzyme generation. Given the mild contributions from the ML aspect, the authors should provide more evaluations on various conditional protein sequence design applications using PLMs and conditional adapters to demonstrate ProCALM's generality and versatility for functional design.


---

[1] Structure-informed language models are protein designers. ICML 2023

[2] Adapting protein language models for structure-conditioned design. BioRxiv 2024

[3] ShapeProt: Top-down Protein Design with 3D Protein Shape Generative Model. BioRxiv 2023

[4] InstructPLM: Aligning Protein Language Models to Follow Protein Structure Instructions.  BioRxiv 2024

**Questions:**

see weaknesses.

---

> ### Author Response · Authors · 2024-11-22
>
> Thank you for your feedback and for pointing us to these highly relevant papers. We have made some significant revisions to our manuscript and uploaded the revision as a new PDF file. While previous studies using adapters have largely focused on conditioning on protein structures, we would like to highlight that ours (ProCALM) is largely the first to explore flexible conditioning with multiple types of inputs (functions), which is an important application for protein engineering. In our revision, we have included a more detailed discussion (and citation) of the previous works above, in the context of our current work **(see Related Work)**.
>
> In terms of technical novelty, we propose and demonstrate the utility of having "parallel adapters" for multiple types of conditioning (e.g.. taxonomy and enzyme family), which has relevant implications for protein generation. To better strengthen our main contributions, we have included a few significant additions in this revision:
>
> 1. To make the broader utility of ProCALM more compelling: We have trained new ProCALM models that utilize textual descriptions to generate proteins more broadly, not just enzymes. After benchmarking against ProteinDT, we found that ProCALM generalizes better and generates higher quality sequences **(see Figure 4)**.
> 2. To provide additional insights on the ML side: We have explored scaling effects for ProCALM by training larger models and found that larger models do not necessarily result in improved performance **(see Figure A.5 and A.6)**.
> 2. To make the evaluation more convincing: We have evaluated the generated sequences with pLDDT and found that our generated sequences have structures with high predicted confidence and are comparable to or better than existing methods **(see Figure 2C and Figure 4E)**.
>
> We hope that these revisions convey how our study is compelling and useful, reiterating that we are the first to explore the use of adapters for conditioning PLMs on function (EC number, natural language, etc.), not just structure. Please let us know if there are further experiments that could be included here to improve our paper.

---

> > ### Comment · Reviewer_5p6F · 2024-11-25
> >
> > I appreciate the authors' efforts in addressing the feedback and revising the manuscript. While I still find the paper somewhat limited in its contributions to the ML perspective, the newly added empirical results on textual conditioning, scaling, and pLDDT enhance the soundness of the work. I accordingly have updated my rating to 5.

---

### Author Response · Authors · 2024-12-03
**Summary of Discussion Period**

Dear Reviewers, ACs, and SACs,

Thank you all for your time and consideration of our work. We summarize our paper here:

Overall, we highlight that our method (ProCALM) is a novel and relevant contribution, as it is the first to explore conditioning protein language models with adapters based on **function**, not just structure. To strengthen this point, in our rebuttal revision, we added a **new task** (natural language-guided conditional generation) and demonstrated that ProCALM outperforms existing methods.

Secondly, we have addressed reviewer concerns regarding the dataset splitting and evaluation. For the latter, we evaluated the structures associated with generated protein sequences, showing that their pLDDT confidences exceed existing methods.

By addressing the reviewer concerns, we raised our scores significantly. There are limitations to our work, such as not always outperforming ZymCTRL and having limited generalization capability. However, we argue that it provides valuable insights to the community and demonstrates a **useful and flexible framework** for conditional generation from protein language models. We hope that you will also find it to be a valuable contribution to the ICLR conference.

Sincerely,

Authors

---

### Meta-Review · Area_Chair_aHgA · 2024-12-22

**Metareview:**

The paper presents ProCALM, an approach for the conditional generation of proteins using adapter tuning with protein language models.
It  demonstrates the ability to generalize to rare and unseen enzyme families and taxonomies, showing potential for broader applications in protein generation. And the use of adapters allows for parameter-efficient training and can accommodate various types of conditioning information. Despite the authors' efforts to highlight novelty, some reviewers still had initial concerns that the approach might seem like an adaptation of existing techniques.

**Additional Comments On Reviewer Discussion:**

In summary, the reviewers had various concerns initially regarding the paper's novelty, performance, evaluation, and OOD generalization. The authors provided detailed responses and revisions, which in most cases led to an improved perception of the paper by the reviewers, with some adjusting their ratings or expressing appreciation for the authors' efforts, although some areas of concern still remained.

---

### Decision · Program_Chairs · 2025-01-22

Reject